# *CYP6B* Subtype Expression Fluctuates in the Great Mormon, *Papilio memnon*, with Changes in the Components of the Host Plants

**DOI:** 10.3390/insects16020159

**Published:** 2025-02-04

**Authors:** Miho Nakano, Takuma Sakamoto, Yoshikazu Kitano, Hidemasa Bono, Hiroko Tabunoki

**Affiliations:** 1Cooperative Major in Advanced Health Science, Graduate School of Bio-Applications and System Engineering, Tokyo University of Agriculture and Technology, Tokyo 183-8509, Japan; nmh.101117@gmail.com; 2Department of Science of Biological Production, Graduate School of Agriculture, Tokyo University of Agriculture and Technology, Tokyo 183-8509, Japan; tsakamoto@go.tuat.ac.jp; 3Department of Applied Biological Science, Tokyo University of Agriculture and Technology, Tokyo 183-8509, Japan; kitayo@cc.tuat.ac.jp; 4Laboratory of Bio-DX, Genome Editing Innovation Center, Hiroshima University, Hiroshima 739-0046, Japan; bonohu@hiroshima-u.ac.jp; 5Laboratory of Genome Informatics, Graduate School of Integrated Sciences for Life, Hiroshima University, Hiroshima 739-0046, Japan

**Keywords:** cytochrome P450, CYP, *CYP6B*, Papilionidae, *Papilio memnon*, insect metabolism, insect CYP, Lepidoptera

## Abstract

Papilionidae larvae metabolise host plant components using a metabolic enzyme, cytochrome P450 (CYP). However, the transcriptomic profile of the *CYP6B* subfamily was insufficiently studied in Rutaceae plant-fed swallowtails. In this study, *CYP6B* mRNA expression was investigated when feeding different kinds of Citrus plants to the Great Mormon (*Papilio memnon*) larvae which use a narrow range of host plants. Consequently, *CYP6B2*, *CYP6B5*, and *CYP6B6* mRNA expressions were changed depending on the kind of Citrus plants provided to the larvae. Furthermore, the chemicals included in the Citrus plants fed to the larvae were different. Our study suggests that phytochemicals cause the alternation of *CYP6B* subtypes mRNA expression, and CYP6B contributes to metabolising phytochemicals in the host plant in *P. memnon* larvae. More studies about the relationship between phytochemicals and the metabolic function of CYP6B will reveal the mechanism of host plant selection in swallowtails.

## 1. Introduction

Herbivorous insects must process toxic components that affect neuroreceptors, nucleic acids, feeding, and digestive systems, which are included in their host plant [1]. Thus, herbivorous insects have some metabolic enzymes, cytochrome P450s (CYPs), glutathione-S transferases (GSTs), UDP-glycosyltransferases (UGTs), and carboxyl/choline esterases (CCEs). These metabolic enzymes can alter the chemical structure of plant components into more soluble and less toxic forms, and then the final metabolites are excreted as frass [2].

*CYP*s are highly conserved among many organisms; these genes are classified into some families, subfamilies, and subtypes according to their similarity with the amino-acid sequence [1,3]. Insect CYP enzymes mainly exist in the midgut, fat bodies, endocrine glands, exocrine glands, some olfactory organs, and epidermal tissues [4].

Lepidoptera has the *CYP6B* subfamily, and it is reported that *CYP6B*’s expression changes when provided with different plant components to polyphagous Lepidoptera. The expression of *CYP6B8*, *CYP6B9*, *CYP6B27*, and *CYP6B28* transcripts is validated in the midgut of the corn earworm (*Helicoverpa zea*) larvae when providing some kinds of plant components [5]. *CYP6B7* is upregulated with xanthotoxin in the cotton bollworm (*Helicoverpa armigera*) larval midgut [6].

In particular, the CYP6B subfamily contributes to the metabolisation of plant components in the swallowtail butterfly [3]. For example, CYP6B1 and CYP6B3 contribute to metabolising xanthotoxin in the black swallowtail (*Papilio polyxenes*, Lepidoptera) [7]. CYP6B4, CYP6B17, and CYP6B21 from the eastern tiger swallowtail (*Papilio glaucus*, Lepidoptera), and CYP6B25 from the Canadian tiger swallowtail (*Papilio canadensis*, Lepidoptera) contribute to metabolising angelicin, trioxsalen, psoralen, xanthotoxin, and bergapten (furanocoumarins) [8]. Furthermore, when swallowtails uptake plant components, *CYP6B*’s mRNA expression is increased. For example, *CYP6B1* and *CYP6B3* mRNA expressions were upregulated by treating xanthotoxin in the *P. polyxenes* larval fat body and midgut [9]. Furthermore, high *CYP6B2* and *CYP6B5* mRNA expressions were induced in the Old World swallowtail’s (*Papilio machaon*, Lepidoptera) larval midgut fed on *Angelica keiskei* than with fennel (*Foeniculum vulgare*, Apiaceae) [10].

The relation of *CYP6B* mRNA expression and plant components is being studied in *P. polyxenes* and *P. machaon*, which feed on Apiaceae plant, as *P. glaucus* and *P. Canadensis* feed on a somewhat wide range of plant families [9,10,11]. Swallowtails feed on plants that belong to a limited family, and they are classified into oligophagous insects according to their food habits [12]. Additionally, swallowtails use specific compounds, including those in the host plant. Thus, they have potential metabolic processes for metabolising them [13]. The swallowtail family of insects makes it easy to study the relationship between food plant components and metabolic enzymes due to the host plant’s limitation.

Many Papilionidae use Rutaceae plants and Apiaceae plants as host plants [12]. The Asian Swallowtail (*Papilio xuthus*) and the Great Mormon (*Papilio memnon*), which use Rutaceae plants as hosts, are evolutionarily distant [14]. In addition, the Asian swallowtail prefers Zanthoxylum species and Citrus species, while the Great Mormon tends to use Citrus species as hosts, i.e., the Great Mormon uses a narrow range of food plants [15]. *CYP6B*s may be involved in the difference in the host plant selection with their evolution, but the relationship between *CYP6B*s and food plant components has yet to be investigated in the Great Mormon.

In this study, the Great Mormon feeds on the following species: Citrus plant, Natsumikan (Japanese summer orange, *Citrus natsudaidai*), Grapefruit (*Citrus* x *paradisi*), Lemon (*Citrus limon*), Shishiyuzu (*Citrus pseudogulgul*), Satsuma orange (*Citrus unshiu*), and Trifoliata orange (*Citrus trifoliata*) (Figure 1). Our previous study showed that the components are different between Natsumikan leaves and Trifoliata orange leaves [16]. Furthermore, Lemon leaves contain more furanocoumarins compared to other Citrus plants, *C. unshiu*, *C. junos*, and *C. depressa* [15]. Unfortunately, the components included in Citrus plants were investigated using fruits within peel and pulp more than leaves. Grapefruit juice contains rich bergamottin and 6′,7′-dihydroxybergamottin which inhibit CYP activity [17,18]. Shishiyuzu fruit has a large size and thick peel compared to the other Citrus plants [19]. Thus, Shishiyuzu might contain different components. Here, this study focused on the expression of *CYP6B*s in the Great Mormon that fed on different host plants and investigated the relationships between host plant components and metabolic enzymes.

## 2. Materials and Methods

### 2.1. Preparation of Insects

*P. memnon* larvae were obtained at the Fuchu campus in the Tokyo University of Agriculture and Technology. The larvae were maintained with the following fresh leaves: Natsumikan, Grapefruit, Lemon, Shishiyuzu, Satsuma orange, or Trifoliata orange at 25 °C under 16 h light/8 h dark cycle. The fresh leaves were obtained at the Fuchu campus in the Tokyo University of Agriculture and Technology.

### 2.2. Preparation of Total RNA

Total RNA was purified using 5th instar larval fat body and midgut using a combination of TRIzol reagent (Thermo Fisher Scientific Inc., Waltham, MA, USA), and PureLink^®^ RNA Extraction Kit (Thermo Fisher Scientific Inc., Waltham, MA, USA) according to the manufacturer’s protocol. The purified total RNA was stored at −80 °C until use.

### 2.3. RNA Sequencing

The quality of the purified total RNAs from Natsumikan-fed larval fat body and midgut was assessed using Agilent TapeStation 2200 (Agilent Technologies, Santana Clara, CA, USA). The cDNA library was constructed from the total RNA with the NEBNext^®^ Poly(A) mRNA Magnetic Isolation Module and NEB NEXT Directional Ultra RNA Library Prep Kit for Illumina^®^ (New England Biolabs, Ipswich, MA, USA) following the manufacturer’s protocol. The RNA sequencing from the libraries (150 bp, paired-end) was conducted with the Illumina NovaSeq6000 platform (Illumina, San Diego, CA, USA).

### 2.4. Transcriptome Analysis

RNA sequencing data from Natsumikan-fed larval fat body and midgut (*n* = 3) trimmed by TrimGalore! Version 0.6.7 (https://www.bioinformatics.babraham.ac.uk/projects/trim_galore/, accessed on 4 December 2024). These trimmed fasta files were mapped on the reference genome of *P. memnon* version 1.0 obtained from NCBI with HISAT2 version 2.2.1 (http://daehwankimlab.github.io/hisat2/, accessed on 4 December 2024). Next, we estimated gene expression using StringTie version 2.1.7 and SAMtools version 1.17 (http://www.htslib.org, accessed on 4 December 2024) [20]. The read count data of transcripts were constructed using prepDE.py (https://github.com/gpertea/stringtie/blob/master/prepDE.py, accessed on 23 January 2025). Differentially expressed genes (DEGs) were analysed using TMM normalization by edgeR package version 4.21 with TCC-GUI version 1.0 (https://github.com/swsoyee/TCC-GUI, accessed on 23 January 2025) in R version 4.4.1 (https://www.r-project.org/, accessed on 23 January 2025). Finally, gene names were put into the transcripts of *P. memnon* using the reference transcripts of *P. xuthus* obtained from NCBI by BLAST version 2.13.0+ (blastn). In total, 26 transcripts of *P. memnon* annotated as *CYP6B* were obtained, and 9 partial transcripts were removed.

### 2.5. Sequence Alignments and Phylogenetic Tree Construction

The base sequences of transcripts in Table 1 were converted to protein sequences using EMBOSS Transeq (https://www.ebi.ac.uk/jdispatcher/st/emboss_transeq, accessed on 17 January 2025). The program ClustalW, in MEGA version 11, was used to align the converted protein sequences of *P. memnon CYP6B*s [21]. A neighbor-joining tree of these aligned sequences was constructed with the bootstrap method by MEGA version 11.

### 2.6. Real-Time Quantitative PCR (RT-qPCR)

The cDNA was synthesised from 500 ng of DNase I (Invitrogen, Van Allen Way, Carlsbad, CA, USA) treated-total RNA, which was extracted from Natsumikan, Grapefruit, Lemon, Shishiyuzu, Satsuma orange, or Trifoliata orange-fed larval fat body or midgut using a PrimeScript™ 1st strand cDNA Synthesis Kit (Takara Co., Ltd., Shiga, Japan) in according to the manufacturer’s instructions. Real-time quantitative PCR (RT–qPCR) was conducted in 20 µL reaction volumes with 0.5 µL of cDNA template, 0.8 µM specific primers (Appendix A), and a KAPA SYBR Fast Qrt–PCR Kit (Nippon Genetics Co., Ltd., Tokyo, Japan). RT–qPCR was performed on a Step One Plus Real-Time PCR System (Applied Biosystems, Foster City, CA, USA) using the Delta–Delta Ct method. RNA expression levels obtained as relative quantification (RQ) values were calculated based on the expression of rpL31 (ribosomal protein L31), which was used as endogenous control. The nucleotide sequences of primers are shown in Appendix A.

### 2.7. Statistical Analysis in RT-qPCR

Dunnett’s test was performed using Delta–Delta Ct values of Natsumikan-fed larval *CYP6B2*, *CYP6B5*, and *CYP6B6* as the control using the General Linear Hypotheses Test contained in multcomp package version 1.4-26 in R version 4.4.1. As *CYP6B2* and *CYP6B5* of the Trifoliata orange-fed larval midgut and fat body, and *CYP6B5* of the Grapefruit- and Shishiyuzu-fed larval fat body contained outliers, these values were removed from the statistical analysis.

### 2.8. Analysis of Components from Citrus Plants

The components contained in 6 kinds of Citrus plants (Natsumikan, Grapefruit, Lemon, Shishiyuzu, Satsuma orange, and Trifoliata orange) were surveyed using TUATinsecta [16]. The data were downloaded on 30 November 2024. The information for the components in Shishiyuzu and Trifoliata orange were not included in this database. Thus, we surveyed using several kinds of literature [19,22,23,24,25,26,27]. Finally, we obtained the information on their components shown in Table 2.

### 2.9. Extraction of Components from Larval Frass

Larval frass fed on Natsumikan, Grapefruit, Lemon, Shishiyuzu, Satsuma orange, or Trifoliata orange, Grapefruit leaves, and Trifoliata orange leaves, which were collected and stocked at −20 °C until use. Freeze-dried frass by lyophilizer (VD-250F, TAITEC Co., Ltd., Saitama, Japan) was immersed into chloroform (FUJIFILM Wako, Osaka, Japan) overnight for extraction. Chloroform extracts were filtrated, and the left frass was further extracted using chloroform three times. The chloroform extracts were concentrated using a rotary evaporator (N-1110 N, Tokyo Scientific Instruments Co., Ltd., Tokyo, Japan). The concentrated chloroform extracts were collected in brown vials and stored at −20 °C until use.

### 2.10. Comparison of Components by Thin-Layer Chromatography (TLC)

The chloroform extracts from each larval frass, Grapefruit leaves, Trifoliata orange leaves, and bergamottin standard purchased from Sigma-Aldrich (St. Louis, MO, USA) were dissolved with chloroform, and they were spotted on a TLC plate (TLC aluminium sheets, Silica gel 60 F_254_, Merck^®^, Darmstadt, Germany) using glass capillaries. The components included in each extract were separated by the mixture of chloroform–methanol (10:1, *v*/*v*). The spots were detected using UVA (366 nm) and UVC (256 nm).

## 3. Results

### 3.1. Transcriptome Analysis of the Fat Body and the Midgut for Detecting CYP6B in Natsumikan-Fed P. memnon Larvae

We investigated the subtype of *CYP6B* transcripts using the midgut and fat body transcriptome using RNA sequencing analysis. We found 27 *CYP6B1*-*7* transcripts expressed in larval fat body and midgut according to the annotation referencing six kinds of the *CYP6B* subtypes in *P. xuthus* (Appendix A). *CYP6B1*, *CYP6B2*, *CYP6B4*, *CYP6B5*, and *CYP6B6* transcripts were expressed in both the midgut and fat body except *CYP6B7* (MSTRG.7509.1). In particular, *CYP6B2* (MSTRG.2041.1, MSTRG.2043.1 and MSTRG.2636.1), *CYP6B4* (MSTRG.10155.1), *CYP6B5* (MSTRG.8816.1 and MSTRG.10157.1), and *CYP6B6* (MSTRG.2039.1 and MSTRG.2040.1) were expressed in the midgut more than ten times higher than in the fat body (Table 1). According to the phylogenetic tree constructed using protein sequences of *CYP6B*s in Table 1, these *CYP6B*s were mainly divided into 2 groups (Group 1; from MSTRG.10155.1 to MSTRG.2040.1, and Group2; from MSTRG.10260.1 to MSTRG.8816.1) based on the bootstrap probability (Appendix A). While Group 1 contained seven kinds of *CYP6B*s, Group 2 contained three kinds of *CYP6B*s, *CYP6B2*, *CYP6B4*, and *CYP6B5*. *CYP6B4*s in Group 1 (MSTRG.10155.1 and MSTRG.10155.2) were splicing variants. Furthermore, nine *CYP6B*s in Table 1 were differentially expressed genes (DEGs) between the midgut and the fat body (Appendix A).

### 3.2. Comparison of CYP6Bs Expression in the Fat Body and the Midgut of Larvae Fed with Different Host Plant

The *CYP6B*s belonging to Group 1 contained divergent *CYP6B*s whose amino acid sequences were similar. The differences in amino acid sequence in CYP6B causes structural and functional divergence [28]. Thus, we considered that the metabolic function of CYP6Bs in Group 1 might be similar. CYP6Bs involved in metabolising host plant components tend to be more highly expressed in the larval midgut than in the larval fat body [9,10]. Therefore, we examined the expression of *CYP6B2* (MSTRG.2636.1), *CYP6B5* (MSTRG.10157.1), and *CYP6B6* (MSTRG.2040.1) transcripts in the midgut and fat body. Because they belonged to Group 1, they were DEGs which were more highly expressed in the midgut than in the fat body, and they were expressed enough in the midgut (Appendix A and Table 1).

*CYP6B2*, *CYP6B5*, and *CYP6B6* mRNA expression were compared among individuals provided with six kinds of Citrus plants, Natsumikan, Grapefruit, Lemon, Shishiyuzu, Satsuma orange, and Trifoliata orange, in larval fat body and midgut by RT-qPCR. There were no differences in the expression levels of the *CYP6B2* and *CYP6B6* mRNA in the larval fat body fed with Natsumikan, Grapefruit, Lemon, Shishiyuzu, and Satsuma orange (Figure 2a,c). In addition, the expression level of the *CYP6B5* mRNA showed little change in the larval fat body fed on Natsumikan, Lemon, and Shishiyuzu. However, the expression level of this gene tended to decrease in Grapefruit-fed larval fat body, and increase in Satsuma orange-fed larval fat body (Figure 2b). Furthermore, the *CYP6B2* and *CYP6B5* mRNA expressions were not detected (Figure 2a,b), and the *CYP6B6* mRNA showed little expression in the Trifoliata-fed larval fat body (Figure 2c).

The expression level of the *CYP6B2* mRNA decreased in the larval midgut fed on Satsuma orange compared to the larval midgut fed on Natsumikan, Grapefruit, Lemon, and Shishiyuzu (Figure 3a). While the *CYP6B5* mRNA in the Grapefruit-fed larval midgut was less expressed than in the Natsumikan-fed larval midgut, the *CYP6B5* mRNA in the Lemon- and Satsuma orange-fed larval midgut was more highly expressed than in the Natsumikan-fed larval midgut (Figure 3b). However, the expression level of the *CYP6B6* mRNA showed little change in the larval midgut fed on Natsumikan, Grapefruit, Lemon, Shishiyuzu, and Satsuma orange (Figure 3c). Interestingly, the *CYP6B2* and *CYP6B5* mRNA expressions were not detected in the Trifoliata-fed larval midgut (Figure 3a,b), and the *CYP6B6* mRNA was slightly expressed in the Trifoliata-fed larval midgut compared to the Natsumikan-fed larval midgut (Figure 3c).

Collectively, the *CYP6B2* mRNA expression in the fat body and the *CYP6B6* mRNA expression in the fat body and midgut might respond to the components commonly included in the tested Citrus plants (except for Trifoliata orange). Furthermore, this expression in the midgut might respond to the components included in Natsumikan, Grapefruit, Lemon, and Shishiyuzu. The *CYP6B5* mRNA expression in the fat body and midgut had the potential to respond to the components which are commonly contained in Natsumikan, Lemon, Shishiyuzu, and Satsuma orange.

### 3.3. Survey of Citrus Plant Phytochemicals Using Database and Literature

Considering that the difference in expression level of the *CYP6B*s mRNA may be caused by chemicals included in the host plant, we surveyed plant chemicals included in the Grapefruit, Lemon, Satsuma orange, and Natsumikan, using TUATinsecta [16], which is a database integrating herbivorous insects, their host plant, and the chemicals including the host plant. Additionally, the chemical information for Shishiyuzu and Trifoliata orange was surveyed using literature [19,22,23,24,25,26,27]. Thus, we found the chemicals in only specific species and those in several species (Table 2). In total, 35 chemicals were included in only Grapefruits, 40 chemicals were included in only Lemon, 32 chemicals were included in only Satsuma oranges, and 61 chemicals were included in only Trifoliata oranges (Table 2). The chemicals included only in the Natsumikan, Shishiyuzu, or common in the host plants were very few (Table 2).

Considering the expression profiles of *CYP6B2*, *CYP6B5*, and *CYP6B6*, we focused on the chemicals contained in Natsumikans, Lemons, and Satsuma oranges, and the chemicals commonly contained in the Citrus plants, except for Trifoliata oranges, used in this study. One hydroxy acid, quinic acid, was included in Natsumikans, Lemons, and Satsuma oranges. Furthermore, two monoterpenes, p-cymene and α-thujene, were commonly included in Grapefruits, Lemons, and Shishiyuzus.

### 3.4. Comparison of the Components Included in the Larval Frass by TLC

Since it was suggested that differences in chemicals included in the host plant might affect the *CYP6B*’s mRNA expression, we investigated the components in the larval frass. Our previous study reported that larval metabolites with altered biological activity compared to host plant components were contained in the chloroform extract from larval frass in *P. machaon* and *P. xuthus* [10,16]. Therefore, the chloroform extracts, which were suggested to contain components metabolised by CYP6Bs, were compared using TLC. The components were extracted using chloroform from the frass of larvae fed on Grapefruits, Lemons, Natsumikans, Shishiyuzus, Satsuma oranges, and Trifoliata oranges, and their metabolites were then compared. The black spot with an Rf value of 0.22 was included in a red spot in the components from the Grapefruit-fed larval frass in Figure 4c, lane 1 (white arrowhead). The spots with Rf values of 0.84 and 0.52 were not detected in Trifoliata orange-fed larval frass in Figure 4a–c, lane 2 (black and white arrowheads). The spots with Rf values of 0.7 (blue) and 0.53 (light blue) were detected in only the Satsuma orange-fed larval frass in Figure 4c, lane 4 (white arrowheads). The Lemon-fed larval frass had no specific spots compared with the other plant-fed larval frass on TLC (Figure 4a–c, lane 6). These results suggested that the components in these larval frass were different.

## 4. Discussion

In this study, we investigated the relationships between the *CYP6B*s mRNA expression and components in *P. memnon* larval fat body and midgut by feeding on different Citrus plants.

We found six kinds of *CYP* transcripts, *CYP6B1*, *CYP6B2*, *CYP6B4*, *CYP6B5*, *CYP6B6* and *CYP6B7*, in *P. memnon* larval fat body and midgut, and focused on *CYP6B5* and *CYP6B6* that were expressed in the midgut than the fat body. We speculated that *CYP6B7* might not have a key role the metabolism of plant components used in this study because the *CYP6B7* mRNA was slightly expressed only in the fat body. *CYP6B1* expression increases in the larval midgut of black swallowtail, *P. polyxenes*, by xanthotoxin which is included in their host, Apiaceae plant, compared to the larval fat body [9]. Thus, it is considered that CYP6B1 metabolises xanthotoxin in the midgut of *P. polyxenes*. In contrast, *CYP6B1* transcript expression was higher in the fat body than in the midgut of *P. memnon* larvae fed on the Citrus plant, Natsumikan, in this study. Furthermore, the leaves of Citrus species contain little furanocoumarin compared to the Skimmia species and Orixa species [15]. Therefore, it was suggested that CYP6B1 from *P. memnon* metabolised Citrus plant components which were different from xanthotoxin and were carried to the larval fat body. CYP family proteins have substrate recognition sites, where the substrate binds and an oxidative reaction is catalysed [4]. The three-dimensional structure is altered depending on the amino acid sequence, and variation of only one amino acid causes the alternation of metabolic function [29,30,31,32]. Thus, a comparison of the structure of CYP6B1 may lead to revealing the difference in the metabolic function between *P. polyxenes* and *P. memnon*.

The expression levels of *CYP6B2* and *CYP6B6* mRNA were not validated in the larval fat body fed on Natsumikans, Grapefruits, Lemons, Shishiyuzus, and Satsuma oranges. However, the expression levels of the *CYP6B2* and *CYP6B6* mRNA tended to decrease in the larval midgut fed with Satsuma oranges. On the other hand, the expression profiles of *CYP6B5* were altered depending on the types of Citrus plants in the *P. mennon* larval fat body and midgut (Figure 2 and Figure 3). As these plants contain many different chemicals (Table 2), *CYP6B5* expression might easily respond to the chemicals in the host plant. Furthermore, the expression level of *CYP6B5* was low in Grapefruit-fed larvae and tended to be high in Satsuma orange-fed larvae (Figure 2b and Figure 3b). Furthermore, it was suggested that Grapefruit- and Satsuma orange-fed larval frass contained different components (Figure 4c). Therefore, CYP6B5 might be involved in the production of these specific components in larval frass fed on Grapefruits and Satsuma oranges. In addition, the *CYP6B2* and *CYP6B5* mRNA expressions were not detected, and the *CYP6B6* mRNA was sightly expressed in the larval midgut and fat body fed with Trifoliata oranges (Figure 2 and Figure 3). The Trifoliate orange chemicals may not be metabolised by these CYP6Bs, therefore, there are some metabolites not included in the Trifoliata orange-fed larval frass.

The contents of furanocoumarins are greatly different among Citrus species [32]; for instance, Lemon and Trifoliata orange leaves contain more furanocoumarins than Satsuma oranges [15]. A furanocoumarin, bergamottin, is contained in Grapefruit juice and inhibits human CYP3A activity [17,18]. We checked for the presence of bergamottin in the chloroform extract of the Grapefruit and Trifoliata orange leaves. However, bergamottin was not detected in these plant extracts on the TLC (Appendix A). This result might suggest that bergamottin did not have a relationship to *CYP6B* transcripts expression in the *P. memnon* larval fat body and midgut. Focusing on the common phytochemicals included in Citrus plants, except for Trifoliata orange, used in this study, we found that p-cymene and α-thujene were included in Grapefruits, Lemons, and Shishiyuzus. Furthermore, Lemons, Natsumikans, and Satsuma oranges commonly contain quinic acid. It is reported that *CYP6B27* and *CYP6B28* are upregulated in the *H. zea* larval midgut by providing chlorogenic acid, which is an esterified compound of quinic acid and cinnamic acid derivatives [5]. Therefore, our results suggested that the expression of *CYP6B*s may respond to specific phytochemicals included in Citrus plants instead of furanocoumarins found in *P. memnon* larvae.

We found that the phytochemicals might be involved in the difference in the expression levels of *CYP6B2*, *CYP6B5*, and *CYP6B6* in the larval fat body and the midgut in this study. According to these results, it was implied that the difference in host plant components affected the *CYP6B* transcript’s expression in the larval fat body and the midgut, and metabolites included in larval frass.

The genomic analysis of *CYP6B*s in Japanese swallowtail insects has reported that differences in host plant components have led to the evolution of diverse roles for *CYP6B*s, affecting its host plant limitation [15].

Although the current study could not clarify the action of each *CYP6B* in the metabolism of phytochemicals, we elucidated that different subtypes of *CYP6B* work according to different phytochemicals.

It is expected that the relationship between the host plants and the function of *CYP6B*s will be revealed by further study, which will reveal the mechanism of host plant selection in Papilionidae.

## Figures and Tables

**Figure 1 insects-16-00159-f001:**
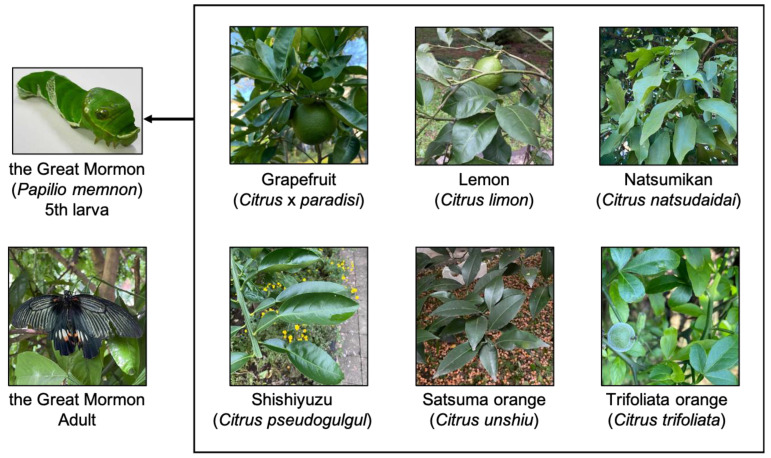
*P. memnon* and six types of host Citrus plants were used in this study.

**Figure 2 insects-16-00159-f002:**
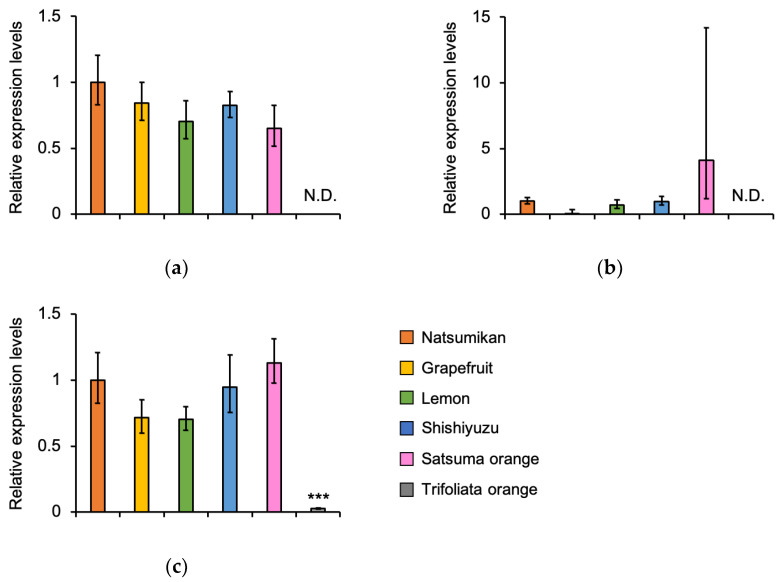
Comparison of *CYP*s expression in the fat body from *P. memnon* larvae fed on six kinds of Citrus plants (Natsumikan, Grapefruit, Lemon, Shishiyuzu, Satsuma orange, and Trifoliata orange) using RT-qPCR. Relative expression levels mean relative quantification (RQ). Bars show RQ minimum and RQ maximum. N.D. means not detected. *** *p* < 0.001, compared with the Natsumikan-fed larval group using Dunnett’s test. (**a**) *CYP6B2* (*n* = 3), (**b**) *CYP6B5* (Natsumikan, Lemon, and Satsuma orange; *n* = 3, Grapefruit and Shishiyuzu; *n* = 2), and (**c**) *CYP6B6* (*n* = 3).

**Figure 3 insects-16-00159-f003:**
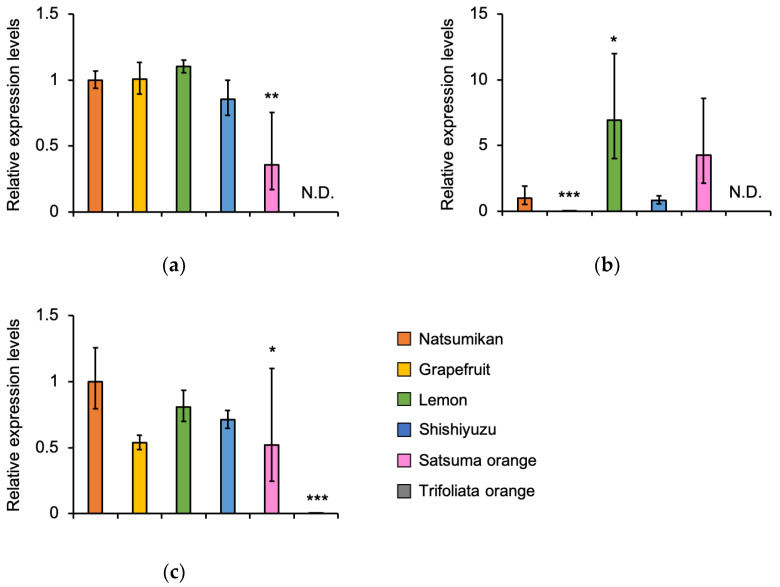
Comparison of *CYP*’s expression in the midgut from *P. memnon* larvae fed on six kinds of Citrus plants (Natsumikan, Grapefruit, Lemon, Shishiyuzu, Satsuma orange, and Trifoliata orange) using RT-qPCR. Relative expression levels mean RQ. Bars show RQ minimum and RQ maximum. N.D. means not detected. *** *p* < 0.001, ** *p* < 0.01, * *p* < 0.05, compared with the Natsumikan-fed larval group using Dunnett’s test. (**a**) *CYP6B2* (*n* = 3), (**b**) *CYP6B5* (*n* = 3), and (**c**) *CYP6B6* (*n* = 3).

**Figure 4 insects-16-00159-f004:**
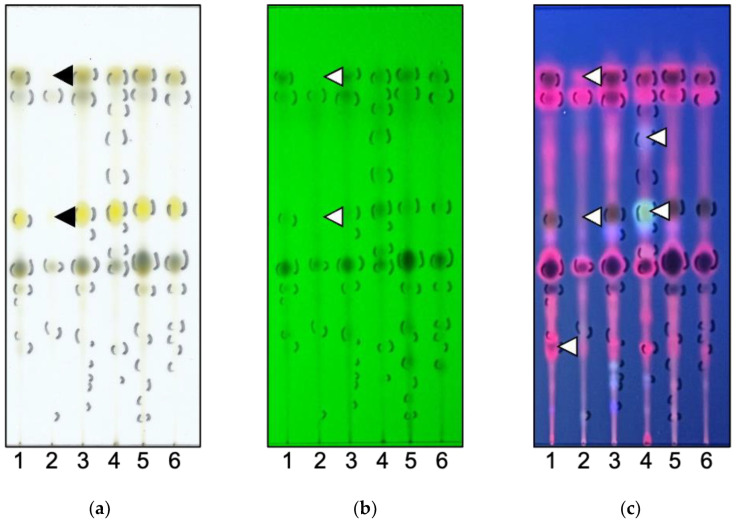
Comparison of components included in chloroform extracts from larval frass by TLC under white light (**a**), UVC at 256 nm (**b**), and UVA at 366 nm (**c**). The marks on the leftside show spots detected by UVC, and those on the rightside show spots detected by UVA. Grapefruit (lane 1), Trifoliata orange (lane 2), Natsumikan (lane 3), Satsuma orange (lane 4), Shishiyuzu (lane 5), and Lemon (lane 6) were used as larval host plants. Black and white arrowheads show the specific spots detected or non-detected in the six types of larval frass.

**Table 1 insects-16-00159-t001:** The annotation of *CYP6B* transcripts and their expression in the midgut and the fat body.

Transcript IDs	Assignment to Corresponding Transcripts	TPM Value		TPM Value
		FB	MG	Ratio *
MSTRG.13945.1	CYP6B1	94.6	0.155	0.01
MSTRG.2041.1	CYP6B2-like	0.0115	35.7	36
MSTRG.2043.1	CYP6B2-like	0.136	28.7	26
MSTRG.2636.1	CYP6B2-like	25.3	414	16
MSTRG.10260.1	CYP6B2-like	7.79	4.44	0.6
MSTRG.10155.1	CYP6B4-like	40.0	531	13
MSTRG.10155.2	CYP6B4-like	78.0	105	1.3
MSTRG.10162.1	CYP6B4-like	24.8	17.3	0.7
MSTRG.8816.1	CYP6B5-like	7.48	1340	159
MSTRG.10157.1	CYP6B5-like	0.292	38.6	31
MSTRG.10158.1	CYP6B5-like	86.0	0.681	0.02
MSTRG.11103.1	CYP6B5-like	10.9	0.123	0.09
MSTRG.11104.1	CYP6B5-like	19.4	0.130	0.06
MSTRG.11105.1	CYP6B5-like	2.18	0.181	0.4
MSTRG.2039.1	CYP6B6-like	0.722	25.3	15
MSTRG.2040.1	CYP6B6-like	115	1370	12
MSTRG.7509.1	CYP6B7-like	0.85	0	0.5

The expression value was calculated by TPM (transcripts per million) value. * TPM value ratio (MG/FB) was calculated using TPM value which added 1.

**Table 2 insects-16-00159-t002:** The number of chemicals included in the Citrus plants.

Citrus Plants	The Number of Chemicals
Grapefruit	35
Lemon	40
Natsumikan	1
Shishiyuzu	4
Satsuma orange	32
Trifoliata orange	61
Grapefruit, Lemon	28
Grapefruit, Trifoliata orange	2
Lemon, Satsuma orange	3
Lemon, Trifoliata orange	1
Natsumikan, Satsuma orange	3
Natsumikan, Trifoliata orange	1
Shishiyuzu, Trifoliata orange	2
Satsuma orange, Trifoliata orange	2
Grapefruit, Lemon, Shishiyuzu	2
Grapefruit, Lemon, Trifoliata orange	4
Natsumikan, Lemon, Satsuma orange	1
Lemon, Shishiyuzu, Trifoliata orange	1
Lemon, Satsuma orange, Trifoliata orange	1
Grapefruit, Lemon, Shishiyuzu, Trifoliata orange	1
Natsumikan, Grapefruit, Lemon, Satsuma orange, Trifoliata orange	1
Grapefruit, Lemon, Shishiyuzu, Satsuma orange, Trifoliata orange	1
Total	227

The chemicals in only specific species and those in several species were counted and shown.

## Data Availability

The RNA sequencing datasets generated and/or analysed during the current study are available in the Sequence Read Archive, DNA Data Bank of Japan repository, under the following accession IDs: fat body groups (DRR622796, DRR622798, DRR622800) and midgut groups (DRR622797, DRR622799, DRR622801).

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
