# Peer review of "CYP6B Subtype Expression Fluctuates in the Great Mormon, Papilio memnon, with Changes in the Components of the Host Plants"

_insects, 2025, doi:10.3390/insects16020159_

Round 1

Reviewer 1 Report

Comments and Suggestions for Authors

Comments to insects-3421000

Cytochrome P-450 (CYP) is one of the metabolic enzymes conserved among organisms to metabolize xenobiotics. The metabolic role and transcriptomic profiles of the CYP6B subfamily have been studied in some Papilionidae insects. This manuscript presents detailed and professional research results about the expression profile of CYP6B in the Great Mormon (Papilio memnon) larvae. However, several issues need to be addressed before it is considered for publication.

Below, I list several points that should be addressed before further consideration for publication.

1.     In terms of this article, it generally lacks innovation and workload. Moreover, this manuscript lacks the storytelling and readability of a scientific paper. Please elaborate on the significance of this study in detail.

2.     The functions of CYP6B genes should be introduced in detail in other insects.

3.     The authors used the RT-qPCR to identify the expressions of 3 kinds of CYP6B genes. What about the other three types of CYP6B genes?

4.     Figures 2 and 3 should add the significance analysis, and the error bar is too big in the treatment group of Satsuma orange.

5.     The functions of CYP6B2, CYP6B5, and CYP6B6 genes need to be studied deeply using RNAi or genome editing systems.

6.     In line 159, italics are required for Latin species names.

In summary, the manuscript requires substantial revision to enhance its clarity, significance, and scientific contribution.

Author Response

January 27, 2025

Response to Reviewer 1

Comment: Cytochrome P-450 (CYP) is one of the metabolic enzymes conserved among organisms to metabolize xenobiotics. The metabolic role and transcriptomic profiles of the CYP6B subfamily have been studied in some Papilionidae insects. This manuscript presents detailed and professional research results about the expression profile of CYP6B in the Great Mormon (Papilio memnon) larvae. However, several issues need to be addressed before it is considered for publication.

Below, I list several points that should be addressed before further consideration for publication.

Response to the reviewer’s comment: Thank you for your comments and suggestions. We have revised our manuscript as you advised. We believe that our revisions satisfy the recommendations you have given.

We have highlighted the revised sections in yellow in the revised manuscript.

Suggestion 1: In terms of this article, it generally lacks innovation and workload. Moreover, this manuscript lacks the storytelling and readability of a scientific paper. Please elaborate on the significance of this study in detail.

Response 1: Thank you for your suggestions. The swallowtail family have few host plants, which facilitates the study of the relationship between the host plant components and their metabolic enzymes. The Great Mormon (Papilio memnon) tends to feed on Citrus species, whereas the Asian swallowtail (P. xuthus) prefers Zanthoxylum and Citrus species. That is, the Great Mormon feeds on a narrower range of host plants. However, the response of CYP6Bs to host plant components has not been fully examined in P. memnon in vitro. Therefore, in this study, the expression profiles of CYP6Bs were investigated in P. memnon larvae fed with different kinds of Citrus plants. We have added this explanation to the Introduction section (Lines 86–90).

Suggestion 2: The functions of CYP6B genes should be introduced in detail in other insects.

Response 2: Thank you for your comment. The CYP6B subfamily are mainly found in Lepidoptera, and CYP6Bs expression changes with the phytochemical composition of plants fed to polyphagous Lepidoptera. The expression of CYP6B8, CYP6B9, CYP6B27, and CYP6B28 transcripts has been validated in the midgut of Helicoverpa zea larvae fed with different kinds of plant components (Li et al., 2002, Insect Molecular Biology, 11, 4). Furthermore, CYP6B7 is upregulated by xanthotoxin in H. armigera larval midgut (Zhou et al., 2010, Pesticide Biochemistry and Physioogy, 97, 3). We have added this explanation to the Introduction section (Lines 60–65).

Suggestion 3: The authors used the RT-qPCR to identify the expressions of 3 kinds of CYP6B genes. What about the other three types of CYP6B genes?

Response 3: Thank you for the point raised in your comment. We selected CYP6B2 (MSTRG.2636.1), CYP6B5 (MSTRG. 10157.1), and CYP6B6 (MSTRG.2040.1) because they were classified under one CYP6B group and were included as differentially expressed genes (DEGs), which are more highly expressed in the midgut than in the fat body. Furthermore, they were more highly expressed in P. memnon fed with Natsumikan. This explanation has been added to the Materials and Methods section (Lines 134–139, 144–149) and the Results section (Lines 208–213, 222–227). We reconstructed the primers specific to these CYP6Bs and provide the revised RT-qPCR data in Figures 2 and 3.

While the expression of CYP6B7 mRNA in the midgut and fat body was too low to be quantified by RT-qPCR, CYP6B1 mRNA in the fat body and CYP6B4 mRNA in the midgut and the fat body were highly expressed. Future studies should evaluate the expression of CYP6B1, CYP6B4, CYP6B2, CYP6B5, and CYP6B6 to uncover clues to the relationship between CYP6Bs and the host plant.

Suggestion 4: Figures 2 and 3 should add the significance analysis, and the error bar is too big in the treatment group of Satsuma orange.

Response 4: Thank you for your comments. Statistical analyses included Delta–Delta Ct values and Dunnett’s test to evaluate Natsumikan-fed larval CYP expression. The results are reported in Figures 2 and 3. We realize that the use of relative quantification (RQ) as relative expression levels in Figures 2 and 3 provided insufficient explanation. The bars in the revised figures indicate the maximum and minimum RQ values. This explanation has been added to the legends of Figures 2 and 3 (Lines 241–244, 265–267).

Suggestion 5: The functions of CYP6B2CYP6B5, and CYP6B6 genes need to be studied deeply using RNAi or genome editing systems.

Response 5: Thank you for your advice. We understand that RNAi or genome editing systems are important analytical methods for elucidating specific gene functions. The CYP genes in the Great Mormon have not yet been characterized based on the results of the feeding experiment. This study is the first to demonstrate the relationships between CYP6B subfamily molecules and host plants of the Great Mormon. Therefore, we will investigate the function of these CYP6B subfamily molecules using RNAi or genome editing systems in future studies.

Suggestion 6: In line 159, italics are required for Latin species names.

Response 6: Thank you for your comment. We have applied the naming conventions for scientific names in lines 198–199.

Reviewer 2 Report

Comments and Suggestions for Authors

Papilio butterflies have been well studied to understand the relationships between insect herbivores and their host plants. Although the majority of Papilio utilizes Rutaceae plants, researchers have mainly focused on Apiaceae-feeders.

              This study analyzed expression patterns of CYP6B genes in response to six different Citrus cultivars in a Rutaceae-feeding Papilio species, P. memnon, potentially providing some insights into the molecular basis of the host plant adaptation in Papilio butterflies. However, I think there are several problems in the present manuscript.

Major points

1) Some studies have already reported the expression profiles of CYP6Bs in Rutaceae plant-feeding Papilio.

As the authors mentioned, Papilio's CYP6B responses to phytochemicals in host plants have been extensively investigated focusing on Apiaceae plant-feeding species mainly performed by Berenbaum and colleagues. Recently, however, some studies were performed using Rutaceae-feeders: Li et al. (2015, Nat. Commun., 6, 8212) pointed out the similarity and difference in CYP6B profiles between Apiaceae-feeding P. machaon and Rutaceae-feeding P. xuthus; Sato et al. (2020, PeerJ, 8, e10625) reported overall CYP6B expression patterns among eight Papilio species inhabiting in Japan, including P. memnon; Miyashita et al. (2024, J. Exp. Biol., jeb247791) showed contribution of furanocoumarin-responsive CYP6B enzymes to furanocoumarin tolerance of P. xuthus larvae. The authors should appropriately clarify purposes, key questions, and novel points of the present study, with description of differences between this study and the previous reports. Additionally, I guess that because of P. memnon's uniquely narrow host plant range (specialist for Citrus plants) the authors decided to use this species. But I cannot find any reason why they focused on P. memnon.

2) What are the criteria for choice of these six Citrus cultivars?

Citrus cultivars are highly diversified, and thus I suppose that wild P. memnon could utilize other cultivars than described in this study. Do these six cultivars reflect taxonomy, or the authors simply use popular cultivars in Japan?

3) Gene or transcript?

Please describe clearly the relationships among the transcripts which are annotated as the same "CYP6B~ -like" in P. memnon. For example, in table 1, I can see seven transcripts annotated as "CYP6B2-like". Are they derived from an identical gene (splicing variants), or different genes belonging in CYP6B2 subgroup? Since approximately thirty CYP6B genes are located on the genome of P. xuthus and classified into CYP6B1–7 subgroups, I (and maybe some readers) am very confused. If the former is correct, please note that gene specific primers for RT-qPCR appropriately amplify all of the variants. If the latter is correct, please designate each gene for discrimination. Phylogenetic tree would be helpful for the readers' understanding.

              Furthermore, please list the information of determined CYP6B sequences. I also think that raw sequencing data obtained from RNA-seq analysis should be deposited to public databases.

4) Statistical analysis.

I cannot find any description of statistical analysis, and I feel this is very serious problem of this study. In addition, the authors performed comparison with the expression value of the Grapefruit-fed larvae. Why? If there is no appropriate background to distinguish Grapefruit from other five Citrus cultivars, multiple comparison among six groups should be conducted.

5) Difference on ingredients in Citrus cultivars (and larval frass).

The difference on the phytochemicals and metabolites among the host plants are presented in Table 2 and Fig.4. Although these data are interesting, the authors provide little explanation for how the differences on the phytochemical profiles affect the different CYP6B expression patterns. What is the conclusion of this analysis?

              In RT-qPCR experiments, Trifoliata orange-fed larvae commonly showed no or few expressions of CYP6B transcripts. Therefore, it is plausible that some specific ingredients which are commonly contained in all tested-Citrus cultivars except for Trifoliata orange induce the CYP6B expression. Unfortunately, there is no description of such phytochemicals in neither table 2 nor main text.

6) Expression pattern of CYP6B1.

Based on the transcriptome analysis using Natsumikan-fed larvae, the authors found that the expression level of P. memnon CYP6B1 was biased in fat bodies. This is an opposite pattern against previous findings in the Apiaceae-feeding P. polyxenes performed by Berenbaum and colleagues, and certainly interesting. Importantly, however, the series of Berenbaum's experiments were performed using host plant leaves treated with 0.2% xanthotoxin, whereas Citrus leaves contain extremely low concentrations of furanocoumarins (even in Lemon, which contains relatively higher levels of furanocoumarins among Citrus cultivars, only 0.0005%, based on data from Sato et al. 2020), leading a concern that the different expression patterns of CYP6B1 between P. polyxenes and P. machaon can be caused by the difference on experimental conditions rather than the species differences. Please discuss carefully.

Minor points

1) Line 159: Please describe that Natsumikan-fed larvae were analyzed for transcriptome.

2) Fig.1: The picture of adult P. memnon is not informative. Characteristic shape and markings on hindwing are completely masked by a leaf.

3) Fig.4: Do orange arrows truly show the specific spots detected in Grapefruit-larval frass? I cannot find any spot near the orange arrow in (a), and I can see the spots on not only lane 1 but also lane 4 in (c).

4) Line 278–283: I feel the major finding of this study is unrelated to insecticide tolerance, and recommend removing this paragraph.

Author Response

January 27, 2025

Response to Reviewer 2

Comment: Papilio butterflies have been well studied to understand the relationships between insect herbivores and their host plants. Although the majority of Papilio utilizes Rutaceae plants, researchers have mainly focused on Apiaceae-feeders. This study analyzed expression patterns of CYP6B genes in response to six different Citrus cultivars in a Rutaceae-feeding Papilio species, P. memnon, potentially providing some insights into the molecular basis of the host plant adaptation in Papilio butterflies. However, I think there are several problems in the present manuscript.

Response to the reviewer’s comment: Thank you for your comments and suggestions. We have revised our manuscript as suggested. We believe that our revisions have satisfied your recommendations.

The additions and revisions are highlighted in yellow in the revised manuscript.

Suggestion 1: Some studies have already reported the expression profiles of CYP6Bs in Rutaceae plant-feeding Papilio.

As the authors mentioned, Papilio's CYP6B responses to phytochemicals in host plants have been extensively investigated focusing on Apiaceae plant-feeding species mainly performed by Berenbaum and colleagues. Recently, however, some studies were performed using Rutaceae-feeders: Li et al. (2015, Nat. Commun., 6, 8212) pointed out the similarity and difference in CYP6B profiles between Apiaceae-feeding P. machaon and Rutaceae-feeding P. xuthus; Sato et al. (2020, PeerJ, 8, e10625) reported overall CYP6B expression patterns among eight Papilio species inhabiting in Japan, including P. memnon; Miyashita et al. (2024, J. Exp. Biol., jeb247791) showed contribution of furanocoumarin-responsive CYP6B enzymes to furanocoumarin tolerance of P. xuthus larvae. The authors should appropriately clarify purposes, key questions, and novel points of the present study, with description of differences between this study and the previous reports. Additionally, I guess that because of P. memnon's uniquely narrow host plant range (specialist for Citrus plants) the authors decided to use this species. But I cannot find any reason why they focused on P. memnon.

Response 1: We appreciate the opportunity to clarify this critical point. Although P. xuthus, whose CYPs have been investigated previously, prefers Zanthoxylum and Citrus pieces, whereas P. memnon tends to feed on Citrus species (Sato et al., 2020, PeerJ, 8, e10625). P. memnon also has a narrower range of host plants than P. xuthus; therefore, the responses of CYPs could be examined by feeding P. memnon larvae with different kinds of plants with specific components. Therefore, we expected to determine the role of CYPs that respond to specific components of Citrus plants. However, the response of CYP6Bs expression to host plant ingredients has not yet been fully examined in P. memnon. Therefore, we investigated the expression profiles of CYP6Bs under different types of Citrus plant diets in P. memnon. We have added this explanation to the Introduction section (Lines 86–90).

Suggestion 2: What are the criteria for choice of these six Citrus cultivars?

Citrus cultivars are highly diversified, and thus I suppose that wild P. memnon could utilize other cultivars than described in this study. Do these six cultivars reflect taxonomy, or the authors simply use popular cultivars in Japan?

Response 2: Thank you for your comment. Our selection of Citrus plants was based on the uniqueness of their components. We previously reported that Natsumikan and Trifoliata orange leaves varied in composition, based on data from the TUATinsecta database (Nakane et al., 2020, Sci Rep, 10, 17509). Lemon leaves contain more furanocoumarin than other Citrus plants (Sato et al., 2020, PeerJ, 8, e10625), while grapefruit is rich in bergamottin, which inhibits CYP activity (Dugrand-Judek A et al., 2015, 10, e0142757). Furthermore, the Shishiyuzu fruit is larger and has a thicker peel than the other Citrus plants studied. Therefore, the components of these plants may differ from one another. This information was added in the Introduction section (Lines 94–99). The Citrus plants used in this study are general cultivars found in Japan.

Suggestion 3: Gene or transcript?

Please describe clearly the relationships among the transcripts which are annotated as the same "CYP6B~ -like" in P. memnon. For example, in table 1, I can see seven transcripts annotated as "CYP6B2-like". Are they derived from an identical gene (splicing variants), or different genes belonging in CYP6B2 subgroup? Since approximately thirty CYP6B genes are located on the genome of P. xuthus and classified into CYP6B1–7 subgroups, I (and maybe some readers) am very confused. If the former is correct, please note that gene specific primers for RT-qPCR appropriately amplify all of the variants. If the latter is correct, please designate each gene for discrimination. Phylogenetic tree would be helpful for the readers' understanding.

Furthermore, please list the information of determined CYP6B sequences. I also think that raw sequencing data obtained from RNA-seq analysis should be deposited to public databases.

Response 3: Thank you for pointing this out. We have already deposited our RNA-seq raw data (accession IDs: fat body groups (DRR622796, DRR622798, DRR622800) and midgut groups (DRR622797, DRR622799, DRR622801) in the Sequence Read Archive, DNA Data Bank of Japan (Data Availability Statement; Lines 393–396). We confirmed the sequence of each P. memnon transcript and removed the partial transcript even if the transcripts annotated CYP6B (Table 1). Therefore, we presented information on the annotated transcript of P. xuthus according to the blastn program (Table S1, Lines 385–386). We constructed the phylogenetic tree using the amino acid sequences of these CYP6Bs in Table 1 (Figure S1, Lines 381–382) and found that two CYP6B4s (MSTRG.10155.1 and MSTRG.10155.2) were splicing variants of each other. The other CYP6Bs were different genes entirely. Thus, we redesigned the specific primers based on our new annotations of CYP6B2 (MSTRG.2636.1), CYP6B5 (MSTRG.10157.1), and CYP6B6 (MSTRG.2040.1) and revised the RT-qPCR graphs in Figures 2 and 3 accordingly. Descriptions of the methods and results of blastn and the phylogenetic tree have been added to the Materials and Methods section (Lines 134–139, 141–149), and the Results section (Line 208–213, 222–227). We have included the data on the CYP6B sequences (CYP6Bs_sequences included in the supplementary information) as supplementary information.

Suggestion 4: Statistical analysis.

I cannot find any description of statistical analysis, and I feel this is very serious problem of this study. In addition, the authors performed comparison with the expression value of the Grapefruit-fed larvae. Why? If there is no appropriate background to distinguish Grapefruit from other five Citrus cultivars, multiple comparison among six groups should be conducted.

Response 4: Thank you for pointing out this clarification. We conducted statistical analysis using Delta–Delta Ct values and Dunnett’s test to evaluate CYP mRNA expression in Natsumikan-fed larvae. The significant results are shown in Figures 2 and 3. The methods are described in the Materials and Methods section (Lines 165–170). However, the CYP6B2 and CYP6B5 in Trifoliata orange-fed larval midgut and fat body and CYP6B5 in grapefruit- and Shishiyuzu-fed larval fat body could not be analyzed because the expression values were too small.

Suggestion 5: Difference on ingredients in Citrus cultivars (and larval frass).

The difference on the phytochemicals and metabolites among the host plants are presented in Table 2 and Fig.4. Although these data are interesting, the authors provide little explanation for how the differences on the phytochemical profiles affect the different CYP6B expression patterns. What is the conclusion of this analysis?

In RT-qPCR experiments, Trifoliata orange-fed larvae commonly showed no or few expressions of CYP6B transcripts. Therefore, it is plausible that some specific ingredients which are commonly contained in all tested-Citrus cultivars except for Trifoliata orange induce the CYP6B expression. Unfortunately, there is no description of such phytochemicals in neither table 2 nor main text.

Response 5: Thank you for your suggestion. Compared with CYP6B2 and CYP6B6, the expression level of CYP6B5 was validated depending on the Citrus plant type (Figures 2 and 3), which varied in composition (Table 2). Therefore, CYP6B5 is easily influenced by the chemical composition of the host plant. In particular, the expression CYP6B5 mRNA was low in the grapefruit-fed larvae but tended to be high in the Satsuma orange-fed larvae. Furthermore, the components of larval frass fed with grapefruit and Satsuma orange differed (Figure 4C). Therefore, CYP6B5 might be involved in the production of the specific components of larval frass fed with grapefruit and Satsuma orange. In addition, the expression of CYP6B2, CYP6B5, and CYP6B6 mRNA was not detected or were low in Trifoliata-fed larvae (Figures 2 and 3). Thus, the components of Trifoliata orange may not be metabolized by these CYP6Bs, which explains the absence of some compounds in the Trifoliata orange-fed larval frass (Fig. 4). These explanations have been added to the Results (Lines 301–303, 304–305) and Discussion (Line s335–350) sections.

Trifoliata orange may have a different chemical composition than other Citrus plants, and CYP6B mRNA expression patterns differed between the midgut and fat body of P. memnon. Considering the expression profiles of CYP6Bs, two phytochemicals, p-cymene and α-thujene, which are commonly contained in Grapefruit, Lemon, and Shishiyuzu, might be involved in the expression of CYP6B2 and CYP6B6. Furthermore, one phytochemical, quinic acid, which is commonly found in lemon, Natsumikan, and Satsuma orange (except for Trifoliata orange), might facilitate the expression of CYP6B2, CYP6B5, and CYP6B6. Based on a study on the polyphagous Lepidoptera, Helicoverpa zea, the transcript levels of CYP6B27 and CYP6B28 increase by feeding plant materials containing quinic acid (Li et al., 2002, Insect Molecular Biology, 11, 4). Thus, our results indicate that the expression of CYP6Bs in P. memnon are influenced by phytochemicals contained in Citrus plants except for furanocoumarin. These explanations have been added to the Results (Lines 280–285) and Discussion (Lines 358–365) sections. Furthermore, for ease of understanding, we have changed the term “ingredients” to “plant compounds” in Table 2.

Suggestion 6: Expression pattern of CYP6B1.

Based on the transcriptome analysis using Natsumikan-fed larvae, the authors found that the expression level of P. memnon CYP6B1 was biased in fat bodies. This is an opposite pattern against previous findings in the Apiaceae-feeding P. polyxenes performed by Berenbaum and colleagues, and certainly interesting. Importantly, however, the series of Berenbaum's experiments were performed using host plant leaves treated with 0.2% xanthotoxin, whereas Citrus leaves contain extremely low concentrations of furanocoumarins (even in Lemon, which contains relatively higher levels of furanocoumarins among Citrus cultivars, only 0.0005%, based on data from Sato et al. 2020), leading a concern that the different expression patterns of CYP6B1 between P. polyxenes and P. machaon can be caused by the difference on experimental conditions rather than the species differences. Please discuss carefully.

Response 6: Thank you for your advice. The expression pattern of CYP6B1 observed in our study is opposite that reported by Berenbaum et al. CYP6B1 metabolizes xanthotoxin efficiently, and the transcript is upregulated in the midgut rather than in the fat body in P. polyxenes; thus, xanthotoxin is preferentially metabolized in P. polyxenes larval midgut. However, the leaves of Citrus plants contain little furanocoumarin. Furthermore, CYP6B1 expression was higher in the fat body than in the midgut in the Natsumikan-fed P. memnon larvae in this study. Therefore, we speculated that CYP6B1 in P. memnon metabolizes Citrus plant phytochemicals other than xanthotoxin and were deposited in the larval fat body. The differences in the amino acid sequences influencing the three-dimensional structure and the substrate recognition site in CYP6Bs may contribute to the differences in metabolic function. Future studies should evaluate the structure of CYP6B1 to uncover the differences in metabolic function between P. polyxenes and P. memnon. We have added a comparison of the metabolic function of CYP6B1 between P. polyxenes and P. memnon in the Discussion section (Lines 324–334).

Minor points:

Suggestion 1: Line 159: Please describe that Natsumikan-fed larvae were analyzed for transcriptome.

Response 1: Thank you for your comment. We have revised the title of the section to “Transcriptome analysis of fat body and midgut for detecting CYP6B in Natsumikan-fed P. memnon larvae” (Lines 198–199).

Suggestion 2: Fig.1: The picture of adult P. memnon is not informative. Characteristic shape and markings on hindwing are completely masked by a leaf.

Response 2: Thank you for pointing this out. We have provided a better photo of an adult Great Mormon in Figure1.

Suggestion 3: Fig.4: Do orange arrows truly show the specific spots detected in Grapefruit-larval frass? I cannot find any spot near the orange arrow in (a), and I can see the spots on not only lane 1 but also lane 4 in (c).

Response 3: Thank you for pointing this out. The black spot was included in a red spot in the components from the grapefruit-fed larval frass in lane 1, Figure 4C (white arrowhead). We have added a more comprehensive explanation in the Results section (Lines 298–229).

Suggestion 4: Line 278–283: I feel the major finding of this study is unrelated to insecticide tolerance, and recommend removing this paragraph.

Response 4: Thank you for pointing this out. We have removed the paragraph discussing insecticides.

Round 2

Reviewer 1 Report

Comments and Suggestions for Authors

The authors answer my questions in detailly.

Author Response

January 30, 2025

Response to Reviewer 1

Comment: The authors answer my questions in detailly.

Answer to the reviewer’s comment: We appreciate your suggestions and comments. Our manuscript was blushed up than before.

Reviewer 2 Report

Comments and Suggestions for Authors

Comments to the authors

I think the authors generally well revised the manuscript. But I feel that more modification is still needed in some part, especially the background of this study and the reason for selection of candidate genes for qPCR analysis.

To response 1)

Lines 85–89: As the authors mentioned in the response letter, CYP6Bs of P. xuthus are considerably analyzed in other studies. However, P. xuthus is phylogenetically close to Apiaceae-feeding Papilios, belonging to distinct clade from the majority of Rutaceae-feeding Papilios. I think that addition of these background to introduction will highlight the importance of this study focusing on P. memnon.

To response 2)

Lines 94–98: Furanocoumarin quantity of Citrus cultivars has mainly been analyzed using fruits (peel and pulp), but less focusing on leaves. Therefore, to my knowledge, it is almost impossible to directly compare the quantity of furanocoumarins contained in the leaves among Citrus species. I understand that the authors have no choice but to cite the studies which only focused on several restricted Citrus plants. I suggest such excuse before the sentence "Furthermore, ~".

Lines 94–95 "Furthermore, only lemon leaves contain furanocoumarin among the Citrus plants [14].": Sato et al., 2020 analyzed 5 Citrus cultivars and several furanocoumarins were detected from all tested Citrus plants, although quantities were very low. Thus, "only lemon contain furanocoumarin" is obviously incorrect.

To response 3)

Table1: The authors identified 27 CYP6B transcripts. And now, 17 transcripts are listed in Table 1. Because the authors mentioned that "We confirmed the sequence of each P. memnon transcript and removed the partial transcript even if the transcripts annotated CYP6B" in the response letter to me, I can guess that 10 transcripts (27 – 17 = 10) were removed. However, this explanation does not appear in the main text or legend. The finding that MSTRG10155.1 and 10155.2 were splicing variants also should be described in the main text.

Lines 221–226 "~ because these were classified into a higher bootstrap probability group.": I don't understand why the reliability of monophyly (high bootstrap value) can be treated as the reason for selection of the candidate genes for qPCR. I also wonder that MSTRG8816.1, 2041.1, and 2042.1 are good candidates because they are judged as DEGs in Table S2.

Line 222: CYP6B5 (MSTRG.2636.1) --> CYP6B5 (MSTRG.10157.1)

Author Response

January 30, 2025

Response to Reviewer 2

Comment: I think the authors generally well revised the manuscript. But I feel that more modification is still needed in some part, especially the background of this study and the reason for selection of candidate genes for qPCR analysis.

Answer to the reviewer’s comment: Thank you for your comments and suggestions. We revised our manuscript according to your suggestions. Our manuscript was blushed up than before. We believe our revisions are satisfied with your suggestions.

We highlighted the revised sections in yellow color in our manuscript.

Suggestion 1: Lines 85–89: As the authors mentioned in the response letter, CYP6Bs of P. xuthus are considerably analyzed in other studies. However, P. xuthus is phylogenetically close to Apiaceae-feeding Papilios, belonging to distinct clade from the majority of Rutaceae-feeding Papilios. I think that addition of these background to introduction will highlight the importance of this study focusing on P. memnon.

Answer 1: Thank you for your suggestion. As you pointed out, P. xuthus and P. memnon, which use Rutaceae plants as host, have evolutionarily distant (Allio et al., 2021, Nature Communications, 12, 1). In addition, P. memnon use narrow range of food plant compared to P. xuthus. CYP6Bs might be involved in the difference of host plant selection with their evolution, however, the relation of CYP6Bs and food plant components has not yet to be examined in P. memnon. This explanation was added to Introduction section (Lines 86–93).

Suggestion 2: Lines 94–98: Furanocoumarin quantity of Citrus cultivars has mainly been analyzed using fruits (peel and pulp), but less focusing on leaves. Therefore, to my knowledge, it is almost impossible to directly compare the quantity of furanocoumarins contained in the leaves among Citrusspecies. I understand that the authors have no choice but to cite the studies which only focused on several restricted Citrus plants. I suggest such excuse before the sentence "Furthermore, ~".

Lines 94–95 "Furthermore, only lemon leaves contain furanocoumarin among the Citrus plants [14].": Sato et al., 2020 analyzed 5 Citrus cultivars and several furanocoumarins were detected from all tested Citrus plants, although quantities were very low. Thus, "only lemon contain furanocoumarin" is obviously incorrect.

Answer 2: Thank you for your suggestions. "Our previous study ~ C. unshiu, C. junos, and C. depressa [14] " (Lines 99–102) mentions the components in Citrus plants leaves. Thus, we added "Unfortunately, the components included in Citrus plants were investigated using fruits with in peel and pulp more than leaves." before the sentence of "Grapefruit juice contains ~" (Lines 102–103). In addition, we revised "Grapefruit contains rich bergamottin~" into "Grapefruit juice contains bergamottin~" in Line 103.

We revised the demonstration about the amount of furanocoumarins in Lemon leaves into "Lemon leaves contain more furanocoumarins compared to other Citrus plants, C. unshiu, C. junos, and C. depressa (Sato et al., 2020)" (Lines 100–102).

Suggestion 3: Table1: The authors identified 27 CYP6B transcripts. And now, 17 transcripts are listed in Table 1. Because the authors mentioned that "We confirmed the sequence of each P. memnon transcript and removed the partial transcript even if the transcripts annotated CYP6B" in the response letter to me, I can guess that 10 transcripts (27 – 17 = 10) were removed. However, this explanation does not appear in the main text or legend. The finding that MSTRG10155.1 and 10155.2 were splicing variants also should be described in the main text.

Lines 221–226 "~ because these were classified into a higher bootstrap probability group.": I don't understand why the reliability of monophyly (high bootstrap value) can be treated as the reason for selection of the candidate genes for qPCR. I also wonder that MSTRG8816.1, 2041.1, and 2042.1 are good candidates because they are judged as DEGs in Table S2.

Line 222: CYP6B5 (MSTRG.2636.1) --> CYP6B5 (MSTRG.10157.1)

Answer 3: Thank you for suggestions. MSTRG.2040.1 was duplicated in the first manuscript, therefore, 17 CYP6Bs were shown in Table 1 by removing 9 partial transcripts from 26 transcripts annotated as CYP6B. This explanation was added to Materials and Methods section (Lines 148–149). Furthermore, the explanation that “MSTRG.10155.1 and MSTRG.10155.2 were splicing variants” was added to Results section (Lines 220–221).

We apologize to insufficient explanation about the selection of CYP6Bs for RT-qPCR. We named the group in phylogenetic tree (Figure S1) as Group 1 (MSTRG.10155.1 to MSTRG.2040.1) and Group 2 (MSTRG.10260.1 to MSTRG.8816.1). Group 1 contained divergent CYP6Bs (CYP6B1 to CYP6B7) whose amino acid sequences were similar. The difference of amino acid sequence causes structural and functional divergence in CYP6B (Li et al., PNAS, 2003), thus, we considered the metabolic function of CYP6Bs in Group 1 might be similar. Therefore, we selected MSTRG.2636.1 (CYP6B2), MSTRG.10157.1 (CYP6B5), and MSTRG.2040.1 (CYP6B6) in Group 1 but not MSTRG.8816.1 (CYP6B5) in Group 2 for RT-qPCR. We added this explanation to Results section (Lines 217–220, 233–236, 239–241).

We revised "CYP6B5 (MSTRG.2636.1)" to "CYP6B5 (MSTRG.10157.1)" in Line 238.
